# T2J: Leveraging Developer Bug-Fixing Behaviors to Evaluate and Improve LLM-Based PyTorch-to-JAX Translation

## Abstract

While Large Language Models (LLMs) have shown strong performance in code-to-code translation for widely-used programming languages, their application to PyTorch-to-JAX translation remains challenging. Although both frameworks are implemented in Python, they differ fundamentally in design principles, execution models, and ecosystem maturity, with JAX being relatively new and under-represented in public code repositories. Moreover, the lack of parallel PyTorch-JAX datasets and the limitations of existing evaluation metrics hinder effective cross-framework translation. In this work, we propose T2J , a prompt augmentation framework aimed at improving LLM-based PyTorch-to-JAX translation. First, we construct two PyTorch code datasets, the problem solving code dataset collected from *TorchLeet* (Aroori & Chien, 2025) repository and the Github code dataset collected from *CodeParrot* benchmark (Wolf et al., 2022), leveraging the cheap LLM 4o-mini to generate initial translations. Second, we employ two professional developers to iteratively fix the generated JAX code until it is functionally equivalent to the original PyTorch code, resulting in a curated *fixed-bug dataset* that captures common translation errors and their corresponding fixes. Third, we design augmented prompts that incorporate structured guidance from the fixed-bug dataset to improve translation quality of lightweight LLMs as GPT-4o-mini. Finally, we take advantages of using LLM as a judge and using LLM to measure the scale of each bug fixing step to propose three evaluation metrics for Pytorch-to-JAX code translation: T2J_CodeTrans_Score, T2J_FixCost_Score, and T2J_Comparison_Score. Our results demonstrate that T2J significantly improves GPT-4o-mini performance by up to **10%** in Code-BLEU, **50%** in T2J_FixCost_Score, **1.33 point** in T2J_CodeTrans_Score (as scale of 0-4), and **100%** in T2J_Comparison_Score. T2J's generated code can improve 2.5 faster in running time compared to the baseline's output execution. Replication package is available at: `https://tinyurl.com/yradutma`.

## 1 Introduction

Code translation involves converting a program from one programming language to another while preserving the original functionality. This process is useful for cross-language and cross-domain migration, allowing organizations to transition their code base to more modern languages or to various purposes. It also supports the modernization of legacy systems by re-implementing them in languages that promote greater maintainability and scalability as part of system refactoring efforts . Furthermore, in enterprises that employ multiple programming languages, code translation enhances the productivity of the programmer.

However, recent research works (Pan et al., 2024; Dou et al., 2024) indicate that LLMs-generated programs in the target language continue to encounter various quality problems, including compilation errors or functional inconsistencies. These challenges become even more pronounced in specialized contexts such as domain-specific language translation. For example, (TehraniJamsaz et al., 2024) demonstrates that their transformer-based approach, CodeRosetta, outperforms well-known LLMs in C-to-CUDA translation. Our work, by contrast, focuses on another domain-specific translation problem: translating between different Python libraries—specifically, converting PyTorch code

snippets into their JAX equivalents. Unlike CodeRosetta, which operates on translation between different programming languages, PyTorch-to-JAX translation cannot be reduced to Abstract Syntax Tree transformations. Relying on low-cost LLMs therefore introduces a significant risk of generating poor-quality translations. This risk arises because JAX, a framework designed for parallelization across diverse hardware platforms, is far less familiar to the broader developer community. Consequently, during PyTorch-to-JAX migration, LLMs often struggle to generate correct JAX code due to their limited exposure to JAX, which is a comparatively newer ecosystem than PyTorch.

To address these challenges, we introduce T2J , a in-context code learning and code evaluation framework designed to enhance LLM-based PyTorch-to-JAX code translation by leveraging curated datasets and structured prompting strategies. This framework proceeds in several key stages: first, we construct parallel corpora of Pytorch and JAX corresponding code snippets from established PyTorch datasets, in particular TorchLeet and CodeParrot (Wolf et al., 2022; Aroori & Chien, 2025). Then we employ high-quality GPT models GPT-4o to produce initial JAX translations. Subsequently, professional human developers iteratively refine the translated JAX program to achieve functional equivalence with the original PyTorch input, producing a curated fixed-bug dataset that systematically documents prevalent translation errors. The dataset, called bug fixing dataset, also includes error-by-error fix instructions as multiple fixing steps. Building on this, we design augmented prompts that integrate targeted, structured guidance derived from the fixed-bug dataset. Finally, we evaluate T2J's performance across both datasets using the CodeBLEU (Ren et al., 2020), alongside three novel metrics: T2J_CodeTrans_Score (assessing the usefulness and functional correctness of LLMs using LLMs as a judge), T2J_FixCost_Score (quantifying the effort required for post-translation corrections), and T2J_Comparison_Score (measuring semantic and functional alignment through differential analysis) to provide a comprehensive assessment of translation quality. Our contributions are as follows:

1. The creation of the first fixed-bug dataset specifically for PyTorch-to-JAX code translation, encompassing detailed annotations of error patterns and fixes to facilitate improvements in reliability on LLM code translation.

2. The T2J framework, which innovates prompt augmentation techniques to bridge domain-specific gaps in cross-library code migration problem as Pytorch-to-JAX translation.

3. The evaluation framework for Pytorch-to-JAX translation, which compare the source/ target code with LLM-prompting techniques and fixing cost from error to correct programs by human bug fixing process.

The remainder of this paper is organized as follows: Section 2 reviews related work, Section 3 describe the Motivation Example, Section 4 introduces the components of T2J , Section 5 presents experimental results, and the final sections discuss limitations, conclusion, and future directions.

## 2 RELATED WORK

**Challenges of Code Translation.** One of the important challenges of code translation is to provide a metric for comparison between predicted and expected code. Research works show that just simply comparing source code by traditional textual similarity scores is not efficient Tran et al. (2019). Instead, code metrics that included information of syntactic/ semantic similarity between code snippets have been proposed (Zhou et al., 2023). Another challenge is that collecting parallel corpus for code translation is very expensive and require human effort for verification (Husain et al., 2020). To evaluate unsupervised code translation's output, automated test cases generation approaches have been proposed (Roziere et al., 2022; Peng et al., 2024). Finally, studies show that there are many types of bugs extracted from LLMs' generated code (Dinh et al., 2023; Zhang et al., 2024).

**Machine Learning-based Approaches for Code Translation.** Supervised methods for code translation are typically trained on well-established datasets such as CodeXGLUE (Lu et al., 2021). Among these methods, BERT-based models have proven particularly effective not only for code-to-code translation but also for a wide range of code generation tasks (Guo et al., 2021; 2022; Wang et al., 2023; Ahmad et al., 2021). Unsupervised code translation typically relies on transforming the source code into an intermediate representation (IR), followed by learning to generate target language code from that intermediate form. Szafraniec et al. (2023) proposed *Transcoder-IR*, a system that uses IR as a pivot language to translate between widely-used languages such as Java,

**(a) Input PyTorch Code**

```python
import torch
import torch.nn as nn
import torch.nn.functional as F

# Define model
class SimpleNN(nn.Module):
    def __init__(self):
        super(SimpleNN, self).
            __init__()
        self.fc = nn.Linear(28 *
            28, 10)

    def forward(self, x):
        x = x.view(-1, 28 * 28)
            # Flatten
        return self.fc(x)

# Example
model = SimpleNN()
input_tensor = torch.randn(1, 1,
    28, 28)
output = model(input_tensor)
print(output)
```

**(b) Incorrect JAX code**

```python
import jax
import jax.numpy as jnp
from jax import random

# Initialize parameters
def init_params(key):
    w_key, b_key = random.split(
        key)
    # Wrong shape
    W = random.normal(w_key, (10,
        28*28)) * jnp.sqrt(1 /
        (28*28))
    b = jnp.zeros((10,))
    return {'W': W, 'b': b}

# Forward function
def forward(params, x):
    x = x.reshape(x.shape[0], -1)
    return jnp.dot(x, params['W'
        ]) + params['b']

# Example
key = random.PRNGKey(0)
params = init_params(key)
input_tensor = random.normal(key,
    (1, 1, 28, 28))
output = forward(params,
    input_tensor)
print(output)
```

**(c) Correct JAX Code**

```python
import jax
import jax.numpy as jnp
from jax import random

# Initialize parameters
def init_params(key):
    w_key, b_key = random.split(
        key)
    # Correct shape
    W = random.normal(w_key,
        (28*28, 10)) * jnp.sqrt
        (1 / (28*28))
    b = jnp.zeros((10,))
    return {'W': W, 'b': b}

# Forward function
def forward(params, x):
    x = x.reshape(x.shape[0], -1)
    return jnp.dot(x, params['W'
        ]) + params['b']

# Example
key = random.PRNGKey(0)
params = init_params(key)
input_tensor = random.normal(key,
    (1, 1, 28, 28))
output = forward(params,
    input_tensor)
print(output)
```

Figure 1: Example of PyTorch-to-JAX translation. (a) Input code; (b) Incorrect translation by 4o-mini; (c) Correct code.

Python, and C++. Huang et al. (2023) proposed *Codist*, which adopts a filtered IR to improve the precision of code translation through a process called code distillation. TehraniJamsaz et al. (2024) presented *CodeRosetta*, a framework for unsupervised translation from C to CUDA. Their method exploits the syntactic similarity between the two languages, leveraging abstract syntax trees (ASTs) as the pivot representation to learn structural correspondences. Roziere et al. (2021) emphasized the significance of a pre-training objective based on recovering broken or obfuscated code.

**Large Language Models for Code Translation.** The emergence of large language models (LLMs) capable of addressing questions across multiple domains has significantly benefited research in code translation. (Zhu et al., 2024) highlighted that state-of-the-art LLMs such as CodeLLaMA often produce translations lacking semantic equivalence—referred to as "shallow translations"—relative to the ground truth. To improve translation quality, Mahmud et al. (2024) proposed *AutoParLLM*, a framework to translate C code into OpenMP pragmatics. Their approach integrates Graph Neural Networks (GNNs) into the prompt to guide LLMs toward better output, and they introduced *OMPScore*, a domain-specific metric tailored to evaluate OpenMP code translations. Tong & Zhang (2024) explored the use of LLMs as evaluators in a multi-phase process, involving code analysis, summarization, and fault localization to assess translated output. Similarly, Ibrahimzada et al. (2025) introduced *AlphaTrans*, a repository-level translation framework that applies LLMs across multiple translation and validation phases. Macedo et al. (2024) proposed *InterTrans*, an LLM-based framework that views code translation as a transitive process that involves intermediate languages.

## 3 MOTIVATION EXAMPLE

An illustration of how low-cost LLM as 4o-mini generate JAX buggy code from Pytorch code can be shown in Figure 1. The input Pytorch program, which is a simple neural network, defines a single-layer feedforward network that flattens a 28×28 input image and applies a linear transformation to produce a 10-dimensional output tensor. The JAX code generated by 4o-mini has an extra function called $init\_params$, while in Pytorch the corresponding function was performed in the construction of the Neural Network object. We can see that the generated code consumed an error in Line 8 (see Figure 1b)), where it incorrectly passed the shape of a linear object defined by the parameter $W$. The correct version of the generated code, shown in Figure 1, requires one modification step to correct the argument passed onto the $random.normal$ JAX API call. This example shows specifc

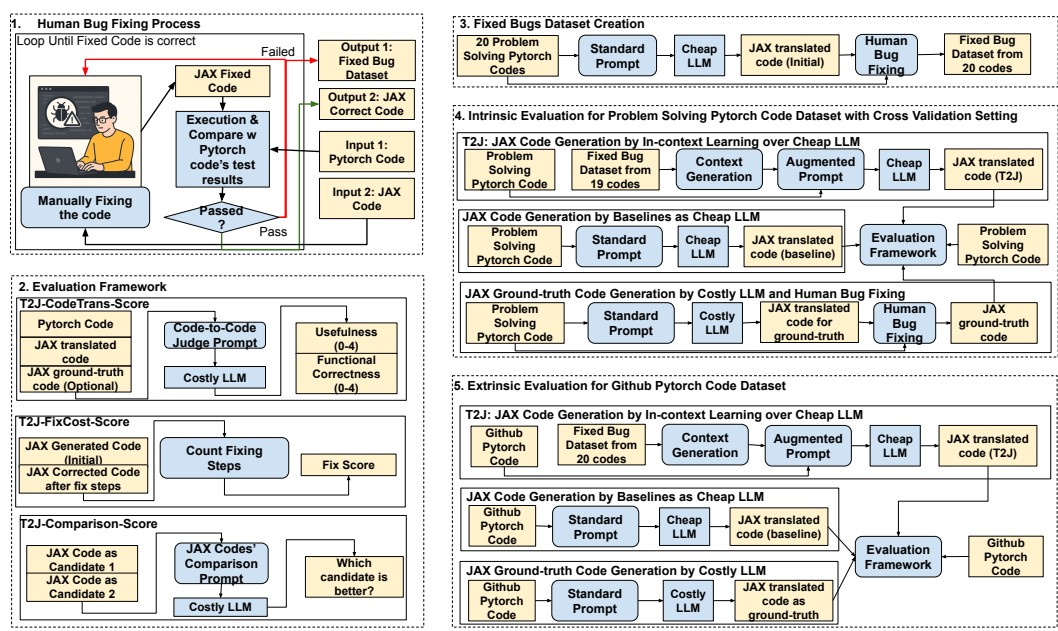

Figure 2: Overview Architecture of T2J

challenge of Pytorch-to-JAX translation that even with this simple code snippet, low-cost LLM still failed to extract the correct code.

# 4 APPROACH

We depict modules of T2J in Figure 2. In the first module, we describe how we hire software developers to check and modify JAX-generated code to ensure it is correct. We also define the definition of correct generated code in our work. The second module introduces our proposed metrics for comparing between predicted and expected generated code, in terms of using LLMs and leveraging the human bug fixing process's output. In the third module, we go in to details about the fixed bug dataset, which we will use for in-context learning with data about fixed bugs as extra context. In the fourth and fifth sections, we describe how T2J performs translation and how we generate the ground-truth dataset for evaluation. We also describe what baseline setting we use for comparison with our proposed pipeline. We call these modules intrinsic evaluation and extrinsic evaluation, and these modules are performed on two different datasets. Next, we discuss about important concepts and design selection we use for T2J .

## 4.1 DESIGN SELECTION

### 4.1.1 SELECTION OF PYTORCH DATASET

We curate datasets of PyTorch code for two tasks. First, we create a dataset of fixed bugs as JAX-generated code from PyTorch input code for our proposed in-context learning process. This process requires the involvement of software developers and LLM for PyTorch-to-JAX code translation. The second task is the evaluation process, where the PyTorch code will be used as input for baseline models or our proposed translation framework to get the output as JAX code snippets for further processing to correct the code and evaluation. Depending on the tasks, we collect the PyTorch dataset from two domains: problem-solving code and general-purpose code.

**Problem-Solving Code Dataset.** We construct a PyTorch dataset based on popular coding interview problems implemented in PyTorch. Owing to the popularity of these problems, we assume that developers can readily verify solutions and fix bugs by consulting existing online resources. For this purpose, we leverage the TorchLeet dataset (Aroori & Chien, 2025) as our problem-solving code corpus. We chose this dataset because each problem is scoped at the file level and because TorchLeet

has been highly ranked by GitHub users. From this dataset, we collect all **20** code snippets, covering three difficulty levels: easy, medium, and hard. All snippets are compilable and runnable using the default test cases included within the dataset.

**GitHub Code Dataset.** To further evaluate our proposed PyTorch-to-JAX translation approach on a broader range of PyTorch code, we consider code snippets drawn from high-quality repositories on GitHub. Specifically, we use the PyTorch subset of the large-scale CodeParrot dataset Wolf et al. (2022) as our second evaluation corpus. From this dataset, we extract **100** PyTorch code snippets. Unlike the Problem-Solving Code dataset, the GitHub PyTorch snippets originate from diverse repositories and are not guaranteed to be directly compilable.

### 4.1.2 SELECTION OF LLMS

**Cheap LLMs.** We define low-cost LLMs as the target models to improve in the PyTorch-to-JAX translation task. These models, referred to as *Cheap LLMs*, can be used without commercial API keys or additional costs. Moreover, we leverage their JAX-generated code to construct the fixed-bug dataset, under the assumption that cheap LLMs produce a higher proportion of buggy code, which can enrich this dataset. For our experiments, we select GPT-4o-mini, the least expensive model offered by OpenAI, as the representative cheap LLM.

**Costly LLMs.** We define high-cost LLMs, referred to as *Costly LLMs*, as the source of ground-truth JAX code for comparison with the translations produced by cheap LLMs. We employ the GPT-4o model as the costly LLM. GPT-4o is one of the most widely used models provided by OpenAI.

### 4.1.3 PROMPTS

We define these following types of prompt in our work.

**Standard Prompt.** We define the standard prompt as the basic prompt for translating code snippet from Pytorch to JAX (see Appendix A.1). In this prompt, we define the translation request by annotating the role of LLM as an expert in code translation and provide a basic request that translate from source language as PyTorch to target language as JAX code.

**Augmented Prompt.** We design augmented prompt with following information. First, similar to the standard prompt, the augmented prompt will have information about the role of LLM and the requirement about input and output. Differ from standard prompt, the augmented prompt will specify a hint for LLMs as the list of errors and errors' solution provided by the fixed bug dataset, uploaded in the JSON format as context of prompt. Definitions of each fields in the fixed bug dataset are also included in the prompt content (see Appendix A.2).

**Evaluation Prompt.** We propose two metrics that leverage LLMs for code-to-code translation. From our knowledge, there are yet existing LLM prompt for doing this task. We define a set of prompts, called evaluation prompts (see Appendix A.3), to ask costly LLMs to evaluate source code in different aspects. We will describe in details of these metrics in the next section.

### 4.1.4 TYPES OF EVALUATION

**Intrinsic and Extrinsic Evaluation.** Our evaluation is divided into two parts. The first, referred to as intrinsic evaluation, is conducted on 20 problem-solving PyTorch code snippets using a cross-validation setting. The second, extrinsic evaluation, is performed on 100 samples of PyTorch code collected from GitHub. There are two key differences between these configurations. First, for intrinsic evaluation, the ground-truth JAX code is obtained through human verification and bug-fixing, whereas for extrinsic evaluation, we rely on JAX code generated by a costly LLM for the GitHub dataset.

### 4.2 HUMAN BUG FIXING PROCESS

The core module of T2J is a systematic bug-fixing process designed to improve the reliability of LLM-generated translations from PyTorch to JAX. Starting with Python code written in PyTorch and the corresponding JAX-generated code from LLMs, this process produces a parallel dataset that pairs the original PyTorch snippets with their corresponding corrected JAX translations. To ensure

Table 1: Evaluation preferences and their descriptions for T2J_CodeTrans_Score.

| Preference | Description |
|---|---|
| Usefulness | How useful the JAX code is for replicating or adapting the functionality of a typical PyTorch source code implementation. |
| Functional Correctness | How well the JAX code preserves the behavior of the original PyTorch code. You are to assess whether the JAX code would produce equivalent outputs to the original PyTorch code across possible inputs, even though the PyTorch code is not shown. Consider unit-test-style logic and general expectations of equivalence. |

correctness, we employed two professional software developers with over five years of Python programming experience to analyze and fix the outputs produced by LLMs. The JAX translations are then subjected to careful manual verification: the developers perform multiple rounds of debugging and correction until the translated JAX code passes all test cases and produces results equivalent to the original PyTorch implementation. During this stage, the verifiers execute the translated and fixed code using Python compilers to confirm correctness. At the end of the process, two complementary datasets are created: one containing pairs of PyTorch snippets and their validated JAX counterparts, and another capturing the bugs identified in LLM outputs along with their corresponding fixes. Importantly, this methodology is flexible, as we apply the same procedure to different PyTorch datasets and experiment with different LLMs depending on the objectives of other modules within our framework.

**Verified Code Correctness by Human.** We consider a fixed version as JAX code snippet verified by a human as correct if and only if the JAX-generated code can be compiled, runnable, and returns the same output as its corresponding Pytorch code snippet, given the same test case. In our work, the human bug fixing process was performed in the problem-solving dataset only since its code snippets have test case and can be runnable which we can rely on their execution output for comparison with the JAX generated code.

### 4.3 EVALUATION FRAMEWORK

We use CodeBLEU Ren et al. (2020), a well-known code evaluation metric, as the baseline metric for PyTorch-to-JAX translation. Since CodeBLEU relies on the AST similarity between two code snippets, we assume that it cannot be a good metric for PyTorch-to-JAX translation output comparison, since both the JAX-generated code might be very similar to the ground truth code, such as in Motivation Example 1. Recently, an LLM-based code evaluation metric has been proposed (Zhuo, 2024). In this work, the authors proposed the ICE-score, a metric for natural language to code translation based on usefulness and functional correctness. In our work, we inherit the idea of the ICE-score and leverage our fixing process to propose three metrics for evaluating JAX-generated code. Given the $n$ pairs of Pytorch and JAX codes in the corresponding Pytorch code set P, the predicted code set JAX $J^p$, the ground truth (reference) code set JAX $J^r$ as $(p_i, j_i)$, our proposing metrics will be provided as following.

**T2J_CodeTrans_Score.** Similar to the ICE-Score metric (Zhuo, 2024) for natural language to code translation, we use the GPT-4o model to evaluate the quality of translated code by usefulness and functional correctness. We design prompt, called CodeTrans prompt, with corresponding evaluation criteria and scoring rubric from 0 (lowest) to 4 (highest). CodeTrans also be useable without having the reference code. Thus, we have a set of following metrics: T2J_CodeTrans_Use_Ref (i.e. the metric for usefulness with reference), T2J_CodeTrans_Func_Ref, T2J_CodeTrans_Use_NoRef, T2J_CodeTrans_Func_NoRef. Explanation of two aspects/ preferences is shown in Table 1.

**T2J_FixCost_Score.** We measure the number of fix steps required to have the JAX correct code from input JAX initial translated code from LLM with this equation:

$$\text{T2J\_}FixCost\_Score(J^{before\_fix}, J^{correct}) = \frac{1}{n} \sum_{i=1}^{n} count\_fix\_step(j_i^{before\_fix}, j_i^{correct}) \quad (1)$$

In equation 1, $J^{before\_fix}$ is the set of JAX generated code as input for human verification process, while $J^{correct}$ is the final version of JAX fixed code that is correct, i.e. it can run and returns consistent output with its corresponding PyTorch code given the same input test case.

**T2J_Comparison_Score.** In this metric, we propose a direct comparison between two translation sets $J^1$ and $J^2$ to see which one is closer to the input PyTorch code set $P$, implemented by this equation:

$$\text{T2J\_Comparison\_Score}(J^1, P, J^2) = \frac{1}{n} \sum_{i=1}^{n} \begin{cases} 1, & \text{if is\_better}(j_i^1, j_i^2, p_i) \\ 0, & \text{otherwise} \end{cases} \tag{2}$$

In equation 2, the $\text{is\_better}(j_i^1, j_i^2, p_i)$ function returns 1 if $j_i^1$ is considered as better code than $j_i^2$, given the input prompt for comparison called Comparison prompt (see Appendix A.3). Note, we also include the content of PyTorch code $p_i$ as a required input to help LLM comparing the input with each code candidate. The scale of this score is from 0 to 1.

### 4.4 Fixed Bug Dataset

Given the input of 20 PyTorch samples from the TorchLeet dataset, two professional developers are hired to modify the code snippets generated from the LLM code translation process. The output of this process for each generated code is a set of multiple fix steps. We store this data in JSON array format to be usable as the context for T2J prompting technique. The fixed bug dataset was constructed by our selected cheap LLM 4o-mini. In total, this dataset contains **163** pairs of bugs/ solutions to fix bug.

### 4.5 Intrinsic Evaluation

In the intrinsic evaluation setting, we want to check if we can use knowledge from fixing 19 problem-solving code pairs of Pytorch and JAX correct code to improve the quality of cheap LLM to translate Pytorch code of the remaining sample. We perform this evaluation as a cross validation process over 20 samples of the fixed bug dataset. Next, we compare the output of three following modules.

**Code generation as Baseline.** We perform the code translation process using the cheap LLM 4o-mini. This process leverages the standard prompt (see Appendix A.1) to translate input PyTorch code to JAX-generated code. We consider this configuration as baseline for T2J for comparison.

**T2J 's In-context learning for code generation.** We perform this process with following modules. First, the augmented prompt will be created with input information as the given Pytorch code snippet and the JSON format of other bug-solutions from other code samples of the problem-solving code dataset. Next, through the cheap LLM, the JAX generated code by T2J 's approach is created. This process was done without the need of any fine-tuning steps, which are usually costly.

**Ground-truth JAX code generation.** We take advantage of costly LLM gpt-4o to generate the JAX code, called JAX initial code from given problem-solving code snippet. Next, the human bug fixing process was performed on this JAX initial code to make the JAX corrected code as ground truth. The main different between this process and the fixed bug dataset creation we mentioned earlier is that this module works with costly LLM. Finally, the output of these three modules will be use as the input of evaluation framework along with the PyTorch source code.

### 4.6 Extrinsic Evaluation

There are two main differences between intrinsic evaluation and extrinsic evaluation. First, for this set up we use the code snippets collected from another dataset, called Github dataset. We evaluate on 100 sample code snippets that are at repository level, meaning that the input code snippets, usually collected from single files, are not guarantee to be runnable and having test cases. Thus, we will use automated metrics we propose for the evaluation. Second, the extrinsic evaluation considered the output of costly LLMs, i.e. JAX translated code by this process) as the ground-truth data point.

Table 2: Results with Cheap LLM (gpt-4o-mini) and Costly LLM (gpt-4o).

| Metrics | Intrinsic Evaluation | | Extrinsic Evaluation | |
|---|---|---|---|---|
| | Baseline | T2J | Baseline | T2J |
| CodeBLEU | 0.19 | 0.29 | 0.41 | 0.38 |
| T2J_CodeTrans_Use._Ref | 1.75 | 2.55 | 2.74 | 2.94 |
| T2J_CodeTrans_Func._Ref | 0.35 | 1.3 | 2.43 | 3.02 |
| T2J_CodeTrans_Use._NoRef | 1.60 | 2.45 | 2.81 | 2.98 |
| T2J_CodeTrans_Func._NoRef | 0.70 | 2.15 | 2.74 | 3.37 |
| T2J_FixCost_Score | 163 | 87 | N/A | N/A |
| T2J_Comparison_Score | 0 | 1 | 0.18 | 0.82 |

Table 3: Correlation of other metrics with T2J_FixCost_Score on Fixed Bug Dataset.

| Metric | Correlation with T2J _FixCost_Score | |
|---|---|---|
| | Pearson | Spearman |
| CodeBLEU | 0.04 | 0.2 |
| T2J_CodeTrans_Use._Ref | 0.2 | 0.25 |
| T2J_CodeTrans_Func._Ref | 0.07 | 0.07 |
| T2J_CodeTrans_Use._NoRef | 0.09 | 0.19 |
| T2J_CodeTrans_Func._NoRef | 0.11 | 0.29 |
| T2J_Comparison_Score | NaN | NaN |

## 5 EXPERIMENT

### 5.1 SETUP

**Hardware Configuration.** For the human bug fixing process, software developers work in the Google Colab environment to debug and fix the code. They use Python 3 as a compiler and use one T4 GPU for running all sample code.

**Question-answering for cheap and costly LLMs.** For both cheap LLM (4o-mini) and costly LLM(gpt-4o), we perform the code generation process through the official interface of ChatGPT-pro. Each question will be created solely in a new topic, and from the answer given by ChatGPT's interface, we manually extract the code snippet as JAX-generated code. Other textual explanation in the output will be omitted. To add context to the existing prompt, we use the Upload function provided by ChatGPT's interface to assign a fixed bug dataset as a JSON file to our designed prompt.

**Executing Evaluation Prompts.** We leverage the access on OpenAI gpt-4o models by API key to get the scoring results for T2J_CodeTrans_Score. For T2J_Comparison_Score, we use ChatGPT-pro's interface to ask questions and receive answers. The reason for using ChatGPT-pro's interface instead of using API key is that some tasks of our work require JSON file upload.

### 5.2 RESULT

#### 5.2.1 TRANSLATION ACCURACY

From Figure 2, our pipeline improves the CodeBLEU score to 0.29, representing a 10% relative gain. In terms of T2J_CodeTrans_Use_Ref, the augmented prompt yields an improvement of 0.8 points over the baseline. For functional correctness, T2J achieves an improvement of 0.95 point. Under the no-reference configuration—where the LLM evaluates only by comparing the input and translated code—T2J still delivers gains of 0.85 and 1.15 points for usefulness and functional correctness, respectively, as measured by the T2J_CodeTrans metrics. Regarding the T2J_Comparison_Score, 100% of the translations generated by T2J are judged superior to the baseline outputs. Finally, in terms of fixing cost, T2J enables GPT-4o-mini to produce code requiring only 87 fixing steps—roughly half the effort compared to fixing the baseline JAX outputs.

Table 4: Comparison of running time (seconds) on Intrinsic Evaluation.

| PyTorch | Ground Truth | Baseline | T2J |
|---|---|---|---|
| 1003 | 851 | 1232.9 | 449 |

Table 5: Comparison of human fixing costs between baseline (weak LLM with standard prompt) and JAX code initially generated by a costly LLM.

| Num. of Fixes | Correcting Weak LLM's Code | Correcting Costly LLM's Code |
|---|---|---|
| Minimum | 1 | 0 |
| Maximum | 32 | 12 |
| Mean | 8.15 | 2.77 |
| Median | 5 | 2 |
| Total | 163 | 61 |

The extrinsic evaluation on the GitHub PyTorch code also highlights the superiority of T2J over the baseline in generating precise code. Interestingly, in this setting the baseline approach outperformed T2J by 3% according to CodeBLEU. In terms of the T2J_CodeTrans metrics, our approach achieves improvements of up to 1.2 point in usefulness and 0.6 point in functional correctness.

### 5.2.2 Correlation of Code Translation Metrics vs Human Fixing Cost

From Table 3, we observe that the T2J _CodeTrans_Score metrics show the strongest correlation with T2J _FixCost_Score under both Pearson and Spearman measures. . Overall, all metrics exhibit weak correlation (below 0.3) with fixing cost. One possible reason is that other metrics are continuous, whereas fixing cost is measured as discrete steps.

### 5.2.3 How close JAX-generated code by costly LLM is to being correct

We further analyze the quality of JAX generated code by costly LLM by meassuring the fixing cost to get the JAX correct code by costly LLM as ground-truth code. The result, shows in Table 5, shows that while it requires much less effort for correcting costly LLM's output than baseline's output, it still requires in total 61 fixing steps to get the correct code set.

### 5.2.4 Analysis on Running Time

We analyze the running time of the corrected code from 3 settings for intrinsic evaluation in Table 4. Results show that T2J can provide significant improvement as 2.5 times faster than running the baseline's output. Details of running time can be seen in Appendix B.

## 6 Limitations

First, the current version of T2J has not yet been applied to improving open LLMs, due to budget constraints that limit our ability to hire software professionals for the human bug-fixing process on these models. Second, our measure of fixing cost is currently based only on the number of fixes, whereas in practice each fix may vary in difficulty. To improve this, there is a need of an algorithm that estimates the relative effort of each bug-fixing step. Third, we conducted the human bug-fixing process only on the problem-solving code dataset, which we want to extend this process for other domains. In future versions of T2J , we plan to collect bug datasets from other general domains.

## 7 Conclusion and Future Works

In this work, we show that T2J can achieve significant improvement compared to baselines as original 4o-mini model for PyTorch-to-JAX code translation. In future work, we attempt to apply our approach for newer open-source LLMs and leverage more advance techniques as supervised fine-tuning and direct preference optimization.

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

APPENDIX

# A  PROMPT TEMPLATE

## A.1  STANDARD PROMPT FOR PYTORCH-TO-JAX TRANSLATION

> You are an expert in programming language translation from PyTorch to JAX. In this task, I will give you input as PyTorch code. Please translate this input PyTorch code to JAX code:
> Input Source Code Snippet: {CODE}

Figure 3: Standard Prompt for PyTorch-to-JAX code translation. The prompt in blue shows the immediate query after the prompt. {CODE}is the starting string of the code.

## A.2  AUGMENTED PROMPT FOR PYTORCH-TO-JAX TRANSLATION

> You are an expert in programming language translation from PyTorch to JAX. In this task, I will give you two inputs:
> 1. Pytorch source code.
> 2. A JSON file that contains a dataset of common errors in PyTorch-to-JAX translation by Weak LLM 4o-mini. Each data point contains the following fields:
> - `Example_id`: ID of the source code.
> - `Input_Code`: Source code in Pytorch.
> - `LLM_weak_output`: JAX translated code of `Input_Code` using a weak LLM (4o-mini). `LLM_fix_output`: Fixed JAX code from `LLM_weak_output` by the process of manually check and fix errors conducted by software developers.
> - `Errors`: This is a list of errors that appeared in the process of manually checking and fixing bugs from `LLM_weak` . Each error item has the following labels:
> • `"Error_Code"`: The part of `LLM_weak_output` that caused the error.
> • `"Error"`: the error message returned by compilation.
> • `"Fix_info"`: the textual description of how to fix the error code
> • `"Fixed_Code"`: The fixed code corresponding to the `"Error_Code"` part.
> 3. The data.csv file thich stored possible input when running some examples in the JSON file. Your task is to reason and get the output JAX code from these above inputs. Please note that you can learn the process of error fixing in Torch-to-JAX translation in 2) JSON file. Now I will give you a set of input in the next query.
>
> Input Source Code Snippet: {CODE}

Figure 4: Prompt for Augmenting to the weak LLM. The prompt in blue shows the immediate query after the prompt. {CODE}is the starting string of the code.

## A.3  EVALUATION PROMPT FOR T2J_CODETRANS_SCORE

### A.3.1  FUNCTIONAL CORRECTNESS

See the prompt without reference at Figure 5 and the prompt with reference at Figure 6.

### A.3.2  USEFULNESS

See the prompt without reference at Figure 7 and the prompt with reference at Figure 8.

> You will be given a JAX code snippet that was translated from PyTorch source code. Your task is to rate the snippet on **one metric only**: its **functional correctness**.
>
> Please ensure you read and understand these instructions carefully before reviewing. Refer to this guide as needed during the evaluation process.
>
> Evaluation Criteria:
>
> Functional Correctness (0–4) — How well the JAX code preserves the behavior of the original PyTorch code.
>
> You are to assess whether the JAX code would produce equivalent outputs to the original PyTorch code across possible inputs, even though the PyTorch code is not shown. Consider unit-test-style logic and general expectations of equivalence.
>
> - A score of 0: The translation is completely incorrect and meaningless.
>
> - A score of 4: The translation is fully correct and handles all core functionalities as expected.
>
> Evaluation Steps:
>
> 1. Assume the code was translated from PyTorch and should preserve its logic.
>
> 2. Evaluate whether the JAX code appears complete, meaningful, and implementationally correct based on general expectations for such translations.
>
> 3. Assign a score for functional correctness on a scale from 0 to 4.
>
> Input Source Code in PyTorch:
>
> {SOURCE_CODE}
>
> Translated JAX Code Snippet:
>
> {TRANSLATED_CODE}
>
> Evaluation Form:
>
> Functional Correctness (scores ONLY):

Figure 5: Prompt for Scoring Functional Correctness by T2J _CodeTrans_Func_NoRef

Table 6: Error categories and their counts in Human Bug Fixing dataset.

| Error Main Category | Count |
|---|---|
| **Training loops, training steps, model fitting** | 32 |
| **Other miscellaneous** | 43 |
| **Model definitions, LinearModel classes, encoders/decoders** | 51 |
| **Loss functions, gradient computation, criterion** | 12 |
| **JAX-specific constructs: jit, grad, PRNG, etc.** | 18 |
| **Iteration patterns: for, while, data loops** | 1 |
| **Parameter updates** | 3 |
| **Final layers, return statements, outputs** | 3 |
| **TOTAL** | 163 |

## A.4 EVALUATION PROMPT FOR T2J_COMPARISON_SCORE

The prompt for querying the T2J_Comparison_Score can be seen in Figure 9.

# B ADDITIONAL RESULTS

## B.1 ANALYSIS ON CATEGORIES OF BUGS

We perform a study on the categorization of bugs on the fixed bug dataset as following. First, two software professionals will go through all the bugs and discuss about the categorizations. Second, from this categorization, they go to the dataset's entities for the second time and do the annotation for categories. We summarize the categorization in Table 6. We further classify types of bugs for some categories to sub-categories, shown in Table 7 and Table 8. We upload each case of this categorization process in the replication package.

> You will be given a JAX code snippet that was translated from PyTorch source code. Your task is to rate the snippet on **one metric only**: its **functional correctness**.
>
> Please ensure you read and understand these instructions carefully before reviewing. Refer to this guide as needed during the evaluation process.
>
> Evaluation Criteria: Functional Correctness (0–4) — How well the JAX code preserves the behavior of the original PyTorch code.
>
> You are to assess whether the JAX code would produce equivalent outputs to the original PyTorch code across possible inputs, even though the PyTorch code is not shown. Consider unit-test-style logic and general expectations of equivalence.
>
> - A score of 0: The translation is completely incorrect and meaningless.
> - A score of 4: The translation is fully correct and handles all core functionalities as expected.
>
> Evaluation Steps:
> 1. Assume the code was translated from PyTorch and should preserve its logic.
> 2. Evaluate whether the JAX code appears complete, meaningful, and implementationally correct based on general expectations for such translations.
> 3. Assign a score for functional correctness on a scale from 0 to 4.
>
> Input Source Code in PyTorch:
> {SOURCE_CODE}
> Translated JAX Code Snippet:
> {TRANSLATED_CODE}
> Reference JAX Code Snippet:
> {REFERENCE}
> Evaluation Form:
> Functional Correctness (scores ONLY):

Figure 6: Prompt for Scoring Functional Correctness by T2J _CodeTrans_Func_Ref

Table 7: Error subcategories under **training loops, training steps, and model fitting**.

| Error Subcategory | Count |
|---|---|
| Misc training issues | 8 |
| Improper passing/using `rng_key`/`prng_key` | 6 |
| Epoch in `range(...)` loop issues | 3 |
| Incorrect usage of Flax `TrainState` and `state.apply_gradients` | 3 |
| Incorrect usage of wrappers (e.g., `train_model(...)` / `fit(...)`) | 3 |
| Train steps return only new state/params without loss at epoch level | 3 |
| JIT/static argument handling for training functions | 3 |
| Errors with batches or dataloaders in training | 1 |
| Loop constructs that break vectorization | 1 |
| Optimizer update/apply patterns in the training loop | 1 |

## B.2 RUNNING TIME ANALYSIS

We perform the running process on a T4 GPU for all the code. We set the timeout of program to run as 180 seconds. Results for each sample in the intrinsic evaluation are shown in Table 9.

## C CONFIGURATIONS

For the task that required user interface action with LLMs, we use the default ChatGPT-pro setting for gpt-4o and 4o-mini. Most of the data were created before July 31st, 2025 when 4o-mini was still available on ChatGPT's interface. For task like LLM-based metric calculation, we leverage Open-Router's API[1] to perform the implementation of these tasks. We also report the hyper parameters for querying costly LLMs for code evaluation in the replication packages.

---

[1]https://openrouter.ai/

Your task is to rate the snippet on **one metric only**: its **usefulness** for understanding and reusing the logic of a typical PyTorch implementation.

Please ensure you read and understand these instructions carefully before reviewing. Refer to this guide as needed during the evaluation process.

Evaluation Criteria: Usefulness (0–4) — How useful the JAX code is for replicating or adapting the functionality of a typical PyTorch source code implementation.

- A score of 0: The JAX translated snippet is irrelevant or confusing and does not help at all.
- A score of 1: The JAX translated snippet includes some related elements but is mostly unhelpful.
- A score of 2: The JAX translated snippet is somewhat useful but needs substantial modification.
- A score of 3: The JAX translated snippet is helpful with minor revisions needed.
- A score of 4: The JAX translated snippet is very helpful and covers the intended functionality clearly.

Evaluation Steps:

1. Assume the PyTorch source code performs a well-defined functionality.
2. Determine whether the JAX translated code snippet enables meaningful reuse or guidance toward equivalent implementation.
3. Assign a score for usefulness from 0 to 4.

Input Source Code in PyTorch:

{SOURCE_CODE}

Translated JAX Code Snippet:

{TRANSLATED_CODE}

Evaluation Form:

Usefulness (scores ONLY):

Figure 7: Prompt for Scoring Usefulness by T2J _CodeTrans_Use_NoRef

Table 8: Error subcategories under **other miscellaneous**.

| Error Subcategory | Count |
|---|---|
| Data arrays, tensors, and dataset values (e.g., creating arrays, specifying shapes) | 8 |
| Dot products with parameters (e.g., `params["w"]`) | 5 |
| Initialization, often in class constructors (`__init__`) | 1 |
| Dot products, sums, or nonlinear transforms | 3 |
| Neural network layers and activations (e.g., `nn.relu`, `nn.Dense`, LSTMs, decoders/encoders) | 5 |
| Tensor dimension errors | 2 |
| Generating synthetic data for CSV | 1 |
| Errors with constant declaration (e.g., epoch) | 1 |
| Incomplete functions/placeholders | 17 |

Your task is to rate the snippet on **one metric only**: its **usefulness** for understanding and reusing the logic of a typical PyTorch implementation.

Please ensure you read and understand these instructions carefully before reviewing. Refer to this guide as needed during the evaluation process.

Evaluation Criteria: Usefulness (0–4) — How useful the JAX code is for replicating or adapting the functionality of a typical PyTorch source code implementation.

- A score of 0: The JAX translated snippet is irrelevant or confusing and does not help at all.
- A score of 1: The JAX translated snippet includes some related elements but is mostly unhelpful.
- A score of 2: The JAX translated snippet is somewhat useful but needs substantial modification.
- A score of 3: The JAX translated snippet is helpful with minor revisions needed.
- A score of 4: The JAX translated snippet is very helpful and covers the intended functionality clearly.

Evaluation Steps:
1. Assume the PyTorch source code performs a well-defined functionality.
2. Determine whether the JAX translated code snippet enables meaningful reuse or guidance toward equivalent implementation.
3. Assign a score for usefulness from 0 to 4.

Input Source Code in PyTorch:
{SOURCE_CODE}
Translated JAX Code Snippet:
{TRANSLATED_CODE}
Reference JAX Code Snippet:
{REFERENCE}
Evaluation Form:
Usefulness (scores ONLY):

Figure 8: Prompt for Scoring Usefulness by T2J_CodeTrans_Use_Ref

You are an expert in PyTorch to JAX translation. I provide 3 inputs: 1. PyTorch input code; 2. Translated Code Candidate A; 3. Translated Code Candidate B. Which candidate is a better translation result for this Pytorch code.
Input Pytorch code:
{CODE}
2. Translated Code A:
{TRANSLATE_CODE_A}
3. Translated Code B:
{TRANSLATE_CODE_B}
Please also provide the reason why you consider a candidate better than the other translated code candidate.

Figure 9: Prompt for T2J_Comparison_Score.

Table 9: Example results comparing PyTorch, ground truth, baseline, and T2J (T2J) outputs.

| Example ID | PyTorch | Ground Truth | Baseline | T2J |
|---|---|---|---|---|
| e1 | 6.61 | 12.7 | 9.78 | 8.79 |
| e2 | 10.00 | 60.0 | 24.4 | 3.59 |
| e3 | 8.57 | 17.3 | 4.14 | 5.18 |
| e4 | 8.82 | 22.1 | 21.8 | 5.36 |
| e5 | 8.48 | 35.0 | 35.2 | 6.53 |
| e6 | 13.0 | 24.0 | 3.0 | 13.68 |
| e7 | 8.23 | 5.0 | 90.0 | 14.53 |
| m1 | 17.0 | 26.0 | 180.0 | 15.62 |
| m3 | 180.0 | 180.0 | 4.0 | 1.05 |
| m4 | 180.0 | 18.0 | 180.0 | 29.0 |
| m5 | 102.0 | 57.0 | 180.0 | 19.5 |
| m6 | 18.0 | 6.0 | 61.0 | 4.0 |
| m7 | 78.0 | 65.0 | 47.9 | 62.0 |
| m8 | 240.0 | 19.0 | 11.8 | 10.0 |
| h1 | 0.48 | 4.0 | 2.5 | 10.0 |
| h3 | 97.0 | 7.0 | 44.1 | 19.0 |
| h4 | 31.1 | 180.0 | 180.0 | 15.0 |
| h5 | 19.6 | 180.0 | 180.0 | 42.0 |
| h6 | 0.31 | 9.0 | 4.0 | 180.0 |
| h10 | 0.89 | 14.0 | 7.6 | 2.0 |
| Total | 1028.09 | 941.1 | 1271.22 | 466.83 |

