# OpenReview forum: "T2J: Leveraging Developer Bug-Fixing Behaviors to Evaluate and Improve LLM-Based PyTorch-to-JAX Translation"
_ICLR.cc/2026/Conference — ICLR 2026 Conference Withdrawn Submission_

### Official Review · Reviewer_mZTJ · 2025-10-20

**Soundness:** 2
**Presentation:** 3
**Contribution:** 3
**Rating:** 2
**Confidence:** 3

**Summary:**

This paper introduces the T2J framework, a code translation tool for a same-language, different-library task: converting from PyTorch to JAX. Due to the relative scarcity of JAX code, optimizing through data fine-tuning is difficult. Therefore, this paper uses a prompt augmentation method to improve performance. The process begins with an initial translation by a "cheap" model, which is then manually corrected by human developers. The error and fix information is preserved and used as an augmented prompt for the model. Experiments show that this method improves the model's translation accuracy. Additionally, the study proposes a new set of evaluation metrics to more accurately measure the usability of the translation and the human effort required to fix the code.

**Strengths:**

1.The research area is novel. It addresses the characteristic data scarcity of this domain by employing an optimization method that does not depend on large-scale datasets.

2.The paper is concisely written, and its figures and tables clearly illustrate the key issues.

**Weaknesses:**

1.The T2J_FixCost_Score is simplistically defined as the inverse of the number of fix steps. However, the cost to fix each error is not uniform and can vary significantly. Defining the "fix cost" solely by the number of steps is overly crude.

2.The scale of the evaluation data is too small (only 20 samples), which is insufficient for the results to be credible.

3.Using an LLM to score the translations lacks credibility, as there is no experiment demonstrating the correlation between the LLM's judgments and human preferences.

4.In the extrinsic evaluation, the ground truth was generated by an LLM without manual verification, which raises significant concerns about its quality.

**Questions:**

1.How was the accuracy of the manual annotations verified? The premise of the study is that LLM-generated code is unreliable, thus requiring human annotation. Why then, for some key metrics, was an LLM's judgment still used for the evaluation?

2.Table 3 shows a very low correlation between the T2J_FixCost_Score and the results from the large model's judgment. Does this suggest that at least one of these two types of metrics is inaccurate?

---

### Official Review · Reviewer_LzbH · 2025-10-26

**Soundness:** 1
**Presentation:** 1
**Contribution:** 1
**Rating:** 0
**Confidence:** 4

**Summary:**

This paper introduces T2J, a framework designed to improve LLM-based PyTorch-to-JAX translation by leveraging human bug fixing behaviors. T2J first builds two PyTorch datasets from TorchLeet and CodeParrot. Then, it uses a lightweight LLM (GPT-4o-mini) to generate initial JAX translations, and then employs expert developers to iteratively fix these outputs, creating a curated bug-free dataset. This dataset is used in augmented prompts that guide LLMs using structured bug-fix knowledge. The authors also propose three new evaluation metrics, namely, T2J CodeTrans Score, FixCost Score, and Comparison Score. Experimental results show T2J improves translation accuracy, up to 10% in CodeBLEU and 50% in FixCost.

**Strengths:**

- Empirical effectiveness
- Use of experts in fixing translation bugs

**Weaknesses:**

- Small dataset size
- Presentation of the paper. For instance, the overview figure is very hard to follow
- Relying on CodeBLEU. Why not perform execution-based validation?
- Result analysis is just too short. The text only mentions results from tables. No interpretation!
- No ablation on the impact of each prompts.

**Questions:**

- Why the authors decided to use CodeBLEU? Almost all recent translation work use Pass@K metrics.
- How do you justify the size of the dataset?
- How would you further interpret the results?

---

### Official Review · Reviewer_6rMC · 2025-10-27

**Soundness:** 2
**Presentation:** 3
**Contribution:** 2
**Rating:** 4
**Confidence:** 5

**Summary:**

The authors propose an in context learning approach to reduce the code translation quality gap between cheap LLMs(eg: GPT 4o-mini) to expensive LLMs(eg: GPT 4o) for PyTorch to JAX code translation tasks.

**Strengths:**

- First contribution of bug fixes dataset for PyTorch to JAX translation with 163 code fix in context learning samples

- Introduction of new scoring metrics inspired by ICE score for DSL code translation tasks such as PyTorch to JAX.

- Clear explanation of motivation and the approach taken to solve the research question.

**Weaknesses:**

- Boundary conditions for effectively leveraging T2J are not discussed. For example following paper shows “ICL (in context learning) incurs substantial computational, memory, and storage costs because it involves processing all of the training examples every time a prediction is made” and proposes PEFT (parameter-efficient fine-tuning) as an alternative - https://proceedings.neurips.cc/paper_files/paper/2022/file/0cde695b83bd186c1fd456302888454c-Paper-Conference.pdf. So, including discussion answering following question would help improve this paper
    - How large bug fixes dataset can grow before its no longer cost effective to use cheaper LLMs and switch over to costly LLMs?
    - How large the bug fixes dataset can grow before context window limits are a problem?

- Problem solving code data set is limited to 20 examples and no analysis/justification is provided to show how these 20 examples have a good representation of the common errors made by the cheaper model like GTP 4o-mini for PyTorch to JAX translation task

- T2J approach is a direct application of ICL (in-context learning), novelty is limited to the DSL application.

Minor Text Corrections
- Line 251-252: “We propose two metrics that leverage LLMs for code-to-code translation.
From our knowledge, there are yet existing LLM prompt for doing this task.”
    - Reason: Typo, Clarity. The overall paper talks about 3 metrics but this line mentions two. “there are yet existing” not sure what his means here.

- Line 43-44: “or to various purposes”
    - Reason: This part is vague. Can remove this part

- Line 56: “Relying on low-cost LLMs therefore introduces a significant risk of generating poor-quality translations.”
    - Reason: Clarity/Out of context. The fact that low cost LLMs translation quality is < high cost LLMs is not established at this point in the paper. This fact is highlighted later. Probably its worth noting this point around discussion at line 50

**Questions:**

- Q1: In section 4.4, How are the fix steps counted ? Is this line count of the edits to the generated code from the models?
- Q2: In section 4.4, Are the software developers using LLMs to fix the bugs/code (or) are they JAX experts?

---

### Official Review · Reviewer_eDAj · 2025-11-01

**Soundness:** 2
**Presentation:** 3
**Contribution:** 2
**Rating:** 2
**Confidence:** 5

**Summary:**

The paper proposes a framework, T2J, to improve code translation between the two Python deep learning frameworks PyTorch and JAX. T2J introduces a prompt-augmentation strategy that leverages developer bug-fixing behavior to enhance low-cost LLM translation quality. The authors generate initial translations with GPT-4o-mini and then have two professional developers iteratively correct them. These corrections form a structured dataset of 163 bug–fix pairs. T2J uses this dataset as contextual guidance within prompts for GPT-4o-mini.

To evaluate translation quality, the paper introduces three metrics: the T2J CodeTrans Score (judging usefulness and functional correctness via LLM-as-a-judge), the FixCost Score (number of fixes required for correctness), and the Comparison Score (pairwise code comparison). Experiments show consistent gains: CodeBLEU improves by up to 10%, fixing cost is halved, and execution time improves by 2.5×.

The main finding/conclusion (A) of this work is in my opinion that adding manually-crafted corrections to prompts improves the transpilation quality, at least for the specific case of PyTorch to JAX translation. There are several secondary contributions (B): new datasets, new metrics, and pointing to this specific but important niche of deep learning code translation.

**Strengths:**

-	It offers a clear and modular pipeline combining data curation, human verification, and in-context learning
-	The idea of capturing developer bug-fixing traces as structured supervision is original and provides interpretable insight into model errors
-	New evaluation metrics move beyond syntactic similarity toward a more functional view of translation quality
-	The experiments, though modest in scale, demonstrate tangible improvements
-	The inclusion of full prompt templates and replication package supports reproducibility

**Weaknesses:**

-	The central assumption of using mostly “cheap” LLMs for C2C translation does not make a lot of sense to me. First, the ratio of performance-to-cost of LLMs is progressively increasing, i.e. today’s expensive LLMs will become cheaper or obsolete in 1 or 2 years. Furthermore, the per-hour cost of professional developers is so much higher that it is not worth saving on LLM usage. Already this point questions the utility of at least the evaluation of this submission. The study would be more compelling if framed as improving weaker or smaller models rather than “cheap” ones. Besides, the term “cheap LLM” is ill-defined: “Cheap LLMs, can be used without commercial API keys or additional costs .“ (L229/230) But you still have to pay for your designated “cheap” LLM GPT-4o-mini.
-	Another key question for judging the value of this paper is: How long-term/persistent is your main insight (that incorporating manual correction data into prompts improves translation)? LLMs evolve rapidly, so to claim long-term relevance, the authors should have shown that the method works over some generations of the past and existing LLMs (from older/weaker to newer/stronger). This could allow a projection for the future generations of LLMs. But the paper does not provide such an evaluation.
-	The third important weakness is a rather shallow conceptual contribution. The idea of prompt enhancement (with a special data, like manual corrections, or not) is an established idea. Most of other conceptual proposals (metrics, your pipeline/architecture) are surely useful but do not excel on the creative side. While the paper focuses on a relevant subdomain of code-to-code (C2C) translation, it lacks theoretical novelty beyond engineering integration.
-	This particular form of prompt enhancement/ in-context learning might not be best one (even given the same correction data) but there is no comparison against chain-of-thought, tree-of-thought, or other “reasoning” methods. It is quite likely that more sophisticated prompt-enhancing techniques could further improve the results.
-	How does in-context learning for C2C translation compares to other techniques, especially fine-tuning and knowledge editing (KE)? For example, KE can work with even single and highly targeted updates (here: corrections), so it would be possible to use the manual correction dataset to make a quantitative, side-to-side comparison. The authors should consider this option to complement their quantitative evaluation (see e.g. https://github.com/zjunlp/EasyEdit for implementing this option).
-	Overall, it is not clear how this method generalizes. The approach and evaluation is constrained primarily to GPT-4o-mini; to a particular prompt-enhancing technique; to a specific scenario (PyTorch and JAX); and uses a relatively small data set. These are quite serious problems of the evaluation, reducing the value of this work.
-	It is quite likely that the method A also works for other framework pairs, like e.g. PyTorch and Tensorflow. The authors should at least discuss how these are scenarios are different (or not) and what would be the potential obstacles for the transfer and obtaining similarly good results. Even a qualitative discussion of such transferability would enrich the paper.
-	It is known that LLMs struggle with C2C translation of larger programs (see e.g. https://arxiv.org/abs/2308.03109), also own experiences of the reviewer with hand-crafted Transpilers (AST to AST translation via rules) for deep learning frameworks indicate that real problems occur when large programs should be translated (e.g. a complete Transformer code). How this approach performs in such scenarios? The provided evaluation on (relatively small) code snippets is insufficient for reliable conclusions for the case of larger programs.
-	The term intrinsic vs. extrinsic evaluation is nonstandard in code translation and could be renamed to “small-scale (verified)” vs “large-scale (automatic)” to improve clarity.
-	Grammar and phrasing are occasionally awkward (“a in-context code learning and code evaluation framework”; “Differ from standard prompt”). A careful language revision would improve readability. There is an unnecessary space in L61 (after T2J). Fonts in Fig. 2 are too small.

**Questions:**

-	In L55 you claim that “PyTorch-to-JAX translation cannot be reduced to Abstract Syntax Tree transformations”. Do you have a reference on this claim or how you arrive at it? The reviever’s team has build a rule-based transpiler for PyTorch-to-JAX translation which works via AST transformations and can translate code up to simple transformers.
-	How reproducible are the T2J metrics when evaluated with a different LLM-as-judge (e.g., Claude or Gemini)?
-	Have the authors tested whether incorporating only error categories (rather than full fixed code snippets) as context yields comparable gains?
-	Could T2J generalize to other intra-language translation tasks (e.g., TensorFlow-to-JAX or NumPy-to-Torch)? (See weaknesses)
-	What is the average context length of the augmented prompts, and does it approach the model’s token limit?

---

### Official Review · Reviewer_6GGQ · 2025-11-02

**Soundness:** 2
**Presentation:** 2
**Contribution:** 2
**Rating:** 4
**Confidence:** 3

**Summary:**

The paper introduces T2J, an LLM-based framework for improving automatic code translation from PyTorch to JAX.
The approach includes:
- Creation of a dataset containing corrections made by two human developers to bugs caused by GPT-4o-mini (a “cheap” LLM) in translating PyTorch code into JAX.
- Use of augmented prompts to include information about bugs fixed from the previous dataset, in order to improve the translation of “cheap” LLMs.
- Proposal of three metrics (T2J CodeTrans Score,
T2J FixCost Score, and T2J Comparison Score), in which an LLM is used as a judge, to evaluate translation performance.
It is shown that the approach improves performance (in the used datasets) both on the proposed metrics and on CodeBLEU.

**Strengths:**

- Dataset with bug fixes.
- Improved translation performance for GPT-4o-mini (cost-effective LLM).
- Evaluation with multiple metrics (in addition to CodeBLEU).

**Weaknesses:**

- The correction dataset is small (20 samples) and limited to GPT-4o-mini. The second dataset (100 samples) is also limited in size, considering that it has not been annotated by humans.
- The “fix cost” metric does not take into account the difficulty of each fix (as reported in the “Limitations” section).
- There are no comparisons with other models (both “cheap” and “costly”), and in particular with open-source models.
- The new LLM-based metrics (T2J CodeTrans Score,
T2J FixCost Score, and T2J Comparison Score) lack in-depth analysis and validation against human or alternative LLM evaluations.
- The organization of the paper could be improved to make it clearer. In particular, the two experiments use different procedures, but the results are described together.
- Some information/details are missing.

**Questions:**

- Did the two professional software developers divide up the annotation work, or did they both work on all 20 problems? Could you also provide some more details on how the individual fixes were identified? I can imagine situations where it would be debatable whether there was one or two fixes. Were there any cases of code changes (tentative fixes) that were then reversed?
- Regarding the proposed evaluation metrics, do you think they should only be used for T2J (given that the names begin with T2J_), or can they be useful in general for translation? In any case, I think a consistency study should be carried out to show correlations and agreement between: different LLM judges, human judges, and the new T2J metrics compared to existing metrics.
- Regarding the “cost” of fix steps, was the time taken by programmers for each fix measured? Although this is only a partial measure of the difficulty, I think that in the absence of anything else (the algorithm discussed in section “6 limitations”) it could be useful. Edit measures could also be used.
- Is Figure 2 so “dense” and not divided into several figures for reasons of space, or is this intentional? Personally, I find it difficult to understand, especially given where it is introduced. Furthermore, it may not be clear which are the “starting blocks” for each part.


### Minor suggestions

- Regarding the two code snippet datasets, some quantitative information could be provided (e.g., number of lines of code, etc.).
- Section 4.1.2 mentions “Cheap LLMs, can be used without commercial API keys or additional costs,” but then GPT-4o-mini is used, which requires an API key and costs. This should probably be argued differently.
- Talking about “cheap/costly” models instead of the computational resources required by the model may be somewhat informal. For example, if OpenAI, for some commercial reason, sold GPT-4o at the same price as GPT-4o-mini, would GPT-4o also become a “cheap” model?
- The problem-solving code dataset consists of 20 problems, but when you go to the page referenced at https://github.com/Exorust/TorchLeet, it is not easy to understand which 20 problems/codes are being referred to.
- “From Figure 2” on page 8 probably refers to Table 2.
- I would expect the “5. Experiment” section to describe the experiment (although I actually think there are two experiments), but in reality it is reserved almost exclusively for the results.
- The extrinsic experiment uses the output of GPT-4o (without human intervention) as the ground-truth metrics. I think it would be better to highlight this limitation more (although it can be inferred indirectly from point three in section “6 Limitations”), or better yet, include other “costly” LLMs and/or a human-verified subset.

---

### Note · Authors · 2025-12-03

**Comment:**

We thank the reviewers for their useful comments and suggestions. We will improve this paper for future venues.

**Withdrawal Confirmation:**

I have read and agree with the venue's withdrawal policy on behalf of myself and my co-authors.